# Identification of Bacterial Communities in Laboratory-Adapted *Glyptotendipes tokunagai* and Wild-Stream-Inhabiting *Chironomus flaviplumus*

**DOI:** 10.3390/microorganisms10112107

**Published:** 2022-10-25

**Authors:** Hokyung Song, Won-Seok Kim, Jae-Won Park, Ihn-Sil Kwak

**Affiliations:** 1Subtropical/Tropical Organism Gene Bank, Jeju National University, Jeju 63243, Korea; 2Department of Ocean Integrated Science, Chonnam National University, Yeosu 59626, Korea

**Keywords:** chironomid, microbiome, wild, cultivated, *Glyptotendipes tokunagai*, *Chironomus flaviplumus*, high-throughput sequencing, detoxification

## Abstract

Chironomidae (chironomid) are one of the dominant families in freshwater ecosystems, and they plays an important role in the food web. They have been used as an indicator for water quality assessment, as they are resistant to diverse environmental pollutants. In this study, we identified the microbiomes of two chironomid species to see if there are any endogenous bacterial groups which could contribute to the host survival. The studied species are *Glyptotendipes tokunagai*, a model species cultivated in a laboratory-controlled environment, and *Chironomus flaviplumus* captured in a field stream in Yeosu, Korea. DNAs were extracted from the whole body of the individual species, and the 16S rRNA gene was amplified. The amplified products were sequenced using an Illumina MiSeq platform. The microbiomes of *G. tokunagai* were homogeneous, having 20% of the core amplicon sequence variants overlapping between replicates sampled from different water tanks. In contrast, none of the core amplicon sequence variants overlapped in *C.*
*flaviplumus*. In both chironomid groups, potential symbionts were identified. *Dysgonomonas*, which can degrade complex carbon sources, was found in more than half of the total microbiomes of *G. tokunagai*. *Tyzzerella* and *Dechloromonas*, which have been suggested to detoxify environmental pollutants, were identified in the microbiome of *C.*
*flaviplumus*. This study can help elucidate the life strategies of chironomids in polluted or organic-rich environments.

## 1. Introduction

Chironomidae (chironomids) are one of the most dominant and widely distributed macroinvertebrates found in freshwater ecosystems. Chironomids undergo complete metamorphosis involving four stages: (1) eggs, (2) larva, (3) pupa, and (4) adult [1]. They spend most of their lifetime in water until they become adults and fly up into the atmosphere. Chironomid larvae are one of the key intermediate members of the food web. They feed on microbes such as algae and are eaten by diverse animal species, including fishes, amphibians, and larger insects.

Chironomid larvae have long been used as water quality indicators of contaminated environments because they can survive in organic-rich and oxygen-deficient environments [2,3]. They can also survive in polluted environments with high concentrations of heavy metals, pesticides, and herbicides [4,5]. In these environments, they apply distinctive survival strategies, such as having hemoglobin that allows them to store oxygen efficiently [6,7]. Furthermore, microbes living inside chironomid eggs and larvae likely help chironomids survive in polluted environments. Using 454 pyrosequencing, Senderovich and Halpern (2013) [8] identified bacterial communities inhabiting chironomids and detected bacterial species that can detoxify metals. Through Koch-like postulates, they showed that the survival rate of disinfected larvae is lower than that of the control groups in the presence of toxic lead. Sela et al. (2021) [5] studied the endogenous microbiota of wild and cultivated (captured in wild and cultivated for 8 months) *Chironomus ramosus* larvae in India and examined wild *C. ramosus* microbiome through metagenomic analysis. They found several genes related to the degradation of toxic compounds, detoxification of metals, and resistance to antibiotics and UV radiation.

Although approximately 10,000 chironomid species exist [9], studies on their microbiome are lacking possibly because, unlike mosquitoes, chironomids are nonbiting midges that do not harm humans. Considering the pivotal role of chironomids in freshwater ecosystems and the potential contribution of endogenous microbes in their survival, it is important to study the microbiomes of various chironomid species. 

In this study, we aimed to identify the microbiome of *Glyptotendipes tokunagai* larvae cultivated for several generations under laboratory conditions and the microbiome of wild *Chironomus flaviplumus* captured in Yeon Deung stream, an urban stream in Korea [8]. *G. tokunagai* and *C. flaviplumus* are two important chironomid species, but their microbiomes have not been studied, to the best of our knowledge. *Glyptotendipes tokunagai* is often used as a model species in ecotoxicology studies, as it can be easily cultivated under laboratory conditions [10,11]. *Chironomus flaviplumus* is commonly found in highly polluted sites [12], and it has gained attention from the public since their presence in a tap water purification plant in Korea in 2020 [13]. We further aimed to infer the functional profile of microbial communities through PICRUST2 [14] and discuss how microbial communities and their functions could contribute to host survival. 

## 2. Materials and Methods

### 2.1. Sample Collection

*Glyptotendipes tokunagai* (GT) was cultivated in a laboratory in accordance with the OECD test 233 [15]. GT was reared in an incubator maintained under the following conditions: 20 ± 1 °C, 70% humidity, light–dark cycle of 16:8 h, and light intensity of approximately 500 lx. M4 medium was used as breeding water, and aeration was continuously supplied. The larvae were fed with Tetra-Min (Tetra-Werke, Melle, Germany) at 0.5 mg per larva per day. GT individuals at the 4th larval stage (three replicates, two individuals per replicate) were collected for this study. Each GT replicate was taken from different water tanks. Wild *Chironomus flaviplumus* (CH) at the 4th larval stage (three replicates, two individuals per replicate) were captured from the Yeon Deung (YD) stream located in Yeosu, South Korea (34°44′ N, 127°43′ E), in February 2022. Developmental stage of chironomids was determined based on body length and body color. The body length of the fourth instar larvae selected for this study was in the range of 5.2–11.2 mm, and the body of the individuals showed clear red color. All of the 12 individuals (2 for each replicate, 3 replicates for GT and 3 replicates for CF) were taken on the same day. The surrounding water samples were collected with three replicates (1 L per each) and filtered through filters with a pore size of 0.22 µm to determine the microbiome of the environment of wild *C. flaviplumus* and to identify possible contamination from skin microbiome.

### 2.2. Assessment of Environmental Factors in the Studied Area

Sampling was conducted at one sampling site in the YD stream. Water samples (1 L) were collected thrice. Several water environmental factors were measured to assess water quality. Water temperature (°C), dissolved oxygen (DO, mg/L), specific conductance (SPC, µS/cm), pH, and salinity (PSU) were determined on site by using a portable YSI Professional Plus (Yellow Springs, OH, USA). For nutrients, 1 L samples were filtered onto precombusted 0.7 µm glass fiber filters (GFF) fused to a Nalgene polysulfone reusable bottle top filter. For filtered GFF, chlorophyll a (Chl-a), NO_2_, NO_3_^−^, NH_4_^+^, PO_4_^3−^, total nitrogen (TN), and total phosphorous (TP) were analyzed in duplicate via a standard spectrophotometric method [16]. All parameters were measured using a GENSYS^TM^ visible spectrophotometer (Thermo Scientific, Waltham, MA, USA). Trophic state indices (TSI) for TP and Chl-a were calculated following the method developed by Carson [17].

### 2.3. DNA Extraction and High-Throughput Sequencing

Total DNAs were extracted from the whole body of GT and CF individuals using a DNeasy blood and tissue kit (Qiagen, Hilden, Germany). Before DNA extraction, individuals were washed several times with 1× PBS to prevent cross-contamination from the field and mechanically homogenized using sterile surgical blades. The total DNA from each of the filtered samples was extracted using the same kit in accordance with a modified protocol [18]. Total DNA was sent to Chunlab Inc. (Seoul, Korea) for 16S rRNA gene amplicon sequencing. The first PCR was carried out under the following conditions: 95 °C for 3 min followed by 25 cycles of 95 °C for 30 s, 55 °C for 30 s, and 72 °C for 30 s. The final extension was conducted at 72 °C for 5 min. Sequencing was performed on an Illumina MiSeq platform (2 × 250 paired ends).

### 2.4. Bioinformatics

Sequences were processed in accordance with the “Moving Pictures” tutorial of Qiime2 software [19] (https://docs.qiime2.org/2022.2/tutorials/moving-pictures/, accessed on 1 March 2022). Sequence quality control was performed using a DADA2 plugin with a minimum length cut-off of 220 bp [20]. Sequences were classified using the Silva database v. 138 [21], and the sequences annotated as “Eukaryote”, “Chloroplast”, “Mitochondria”, and “Archaea” were removed for further processing. PICRUST2 v. 2.4.2 was applied to predict the functional profile of the microbial communities [14]. Before PICRUST2 was used, the number of sequences in each sample was normalized with 3793 reads per sample. The raw fastq files have been deposited at the Sequence Read Archive (SRA) under project ID PRJNA880421.

### 2.5. Statistical Analysis

Rarefaction curves were generated using the “rarecurve” function in R package “vegan” [22] with a step size of five. A principal coordinate analysis (PCoA) plot was generated with R “vegan” package based on the Bray–Curtis dissimilarity between samples calculated with square-root-transformed amplicon sequence variant (ASV) abundance data. Venn diagrams of ASVs were generated for each chironomid species using R “ampvis2” package [23]. The ASVs over 10% relative abundance were considered as core ASVs.

## 3. Results

### 3.1. Environmental Conditions of the Studied Sites

The environmental conditions of the studied area are summarized in Table 1. The trophic state index (TSI) was 50.8 when calculated based on the average chlorophyl-a concentration and 75.5 when calculated based on the average total phosphorous concentration, representing an eutrophic condition.

### 3.2. Microbial Community Composition

The rarefaction curves for the chironomid samples were saturated at the subsampling point (3793 reads), although the curves were unsaturated for the filter samples (Figure 1). The PCoA plot showed a clear distinction between the microbiomes of wild and cultivated chironomids (Figure 2). The microbiome of wild *C. flaviplumus* was also separated from the microbial communities of the surrounding water environment (Figure 2). In terms of diversity, the Shannon diversity (α-diversity) and the within-group Bray–Curtis dissimilarity (β-diversity) were the lowest in the cultivated *G. tokunagai* microbiome (Figure 3). About 20% of the core ASVs overlapped between *G. tokunagai* replicates, whereas in the replicates of *C.*
*flaviplumus* none of the core ASVs overlapped (Figure 4).

The three most abundant phyla accounting for more than 85% in all the studied samples were *Proteobacteria*, *Bacteroidota*, and *Firmicutes* (Figure 5). At the genus level, *Dysgonomonas* was found in more than half of all *G. tokunagai* samples and in one of the wild *C. flaviplumus* samples (Figure 6). In two of the wild *C. flaviplumus* samples, *Sphaerotilus* and *Dechloromonas* were dominant, accounting for 20% and 5% each of the total abundance. *Tyzzerella* and *Flavobacterium* were found to be abundant in all three *C. flaviplumus* samples. The average relative abundance of *Dysgonomonas, Sphaerotilus*, and *Tyzzerella* was less than 1% in Yeon Deung samples. In contrast, *Flavobacterium* was found to be abundant (average relative abundance of 6%) in Yeon Deung samples as well. 

### 3.3. Functional Potential of Microbial Communities

Table 2 summarizes the genes related to resistance to antibiotics, toxic metals, and toxic compounds that were identified in our study via PICRUST2 and in *C. ramosus* larval microbiome in Sela et al.’s study (2021) [5] through metagenome sequencing. These included an arsenic resistance gene (arsH), cobalt–zinc–cadmium resistance genes (e.g., czcA and zitB), copper resistnace genes (e.g., cusR, copR, and pcoD genes), multidrug resistance genes (e.g., mdtK and emrE), and a gene related to phenylacetic acid degradation (paaY).

## 4. Discussion

In this study, we used high-throughput sequencing methods to identify the microbiomes of cultivated *G. tokunagai* and wild *C. flaviplumus*. As indicated by the low within-group dissimilarity and highly overlapping core ASVs, the microbiome of *G. tokunagai* was homogeneous suggesting that environmental conditions were controlled well. The Shannon diversity of *G. tokunagai* microbiome was low as well, possibly because they have been cultivated for several generations, which could result in simpler microbial forms that only include key microbes.

At the phylum level, *Proteobacteria*, *Bacteroidota*, and *Firmicutes* were dominant in both groups; this finding was consistent with previous results on chironomid microbiome [5,8,24]. At the genus level, *Dysgonomonas* was observed in more than half of the microbiomes of all three *G. tokunagai* individuals. *Dysgonomonas* is often identified as one of the most dominant genera in the microbiome of insects, including chironomids [5,24,25]. *Dysgonomonas* is an anaerobic bacterial genus that grows on nutrient-rich media and can degrade diverse polysaccharides and complex compounds, such as lignocellulose [26]. In chironomids, they could work as potential symbionts that provide nutrients as simpler forms to their host.

*Sphaerotilus, Tyzzerella*, and *Dechloromonas,* were found to be dominant in *C. flaviplumus. Sphaerotilus* is represented by *Sphaerotilus natans*, which have often been isolated from nutrient-rich environments such as sewage and activated sludge samples [27,28,29]. *Tyzzerella* and *Dechloromonas* are associated with various types of environmental pollutants. The relative abundance of *Tyzzerella* was higher in the gut microbiome of *Achirus lineatus* exposed to a water-accommodated fraction of light crude oil [30] and in the gut microbiome of residents with long-term exposure to heavy metals in a mining area [31]. Species belonging to *Dechloromonas* have a broad range of metabolic capabilities including reduction of perchlorate and oxidization of chlorobenzoate, toluene, and xylene [32,33]. For this reason, they have been suggested as candidate organisms for bioremediation [34]. Possibly, these organisms could aid detoxification of pollutants and provide nutrient sources to their host. *Flavobacterium* was one of the abundant taxa in the microbiome of wild *C. flaviplumus* as well, but it was also abundant in the surrounding environment. This could be due to contamination from the surrounding environment or from skin microbiome. The high abundance of this taxa corresponds with previous findings that show prevalence of *Flavobacterium* in eutrophic lakes [35,36].

Although heavy metal concentration was not measured in this study, a previous study shows a high level of heavy metals in the sediment of Yeon Deung stream, exceeding the limit designated by Korean law, Enforcement decree of the water quality and aquatic ecosystem conservation act [37]. Functional genes inferred by PICRUST2 included gene categories related to resistance to antibiotics, toxic metals (e.g., copper, arsenic, and zinc), and toxic compounds (e.g., quaternary ammonium compound), as found by Sela et al. (2021) [5]. This finding strengthened the hypothesis of Senderovich and Halpern (2013) [8], who described the protective role of chironomid microbiome. The copper resistance gene in *G. tokunagai* microbiome was more abundant than that in *C. flaviplumus* microbiome, possibly because of the water pipe composed of copper, although the copper level in the cultivation environment was not tested in this study.

## 5. Conclusions

In this study, we identified the microbiome of cultivated *G. tokunagai*, which is often used as a model species, and the microbiome of wild *C. flaviplumus*. We found that *G. tokunagai* possessed a homogeneous microbiome where a single genus (*Dysgonomonas*) dominating, possibly because *G. tokunagai* was cultivated in a controlled environment. In both chironomid groups, we found bacterial genera which have ability to degrade complex carbon sources possibly aiding host digestion. *Tyzzerella* and *Dechloromonas* that could detoxify toxic metals/compounds were more abundant in wild *C. flaviplumus* than in *G. tokunagai*, strengthening the hypothesis on the protective role of chironomid microbiome against environmental pollutants. In the future, metatranscriptomics will provide further information to understand the response mechanisms of chironomids against toxic compounds and the symbiotic roles of chironomid microbiome.

## Figures and Tables

**Figure 1 microorganisms-10-02107-f001:**
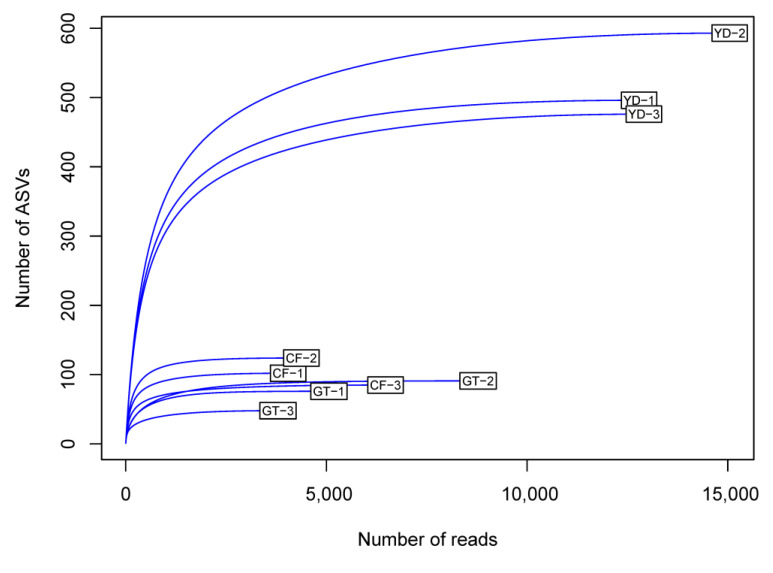
Rarefaction curves for the chironomid samples (GT and CF) and the filter samples (YD). GT: G. to-kunagai; CF: C. flaviplumus; YD: Yeon Deung water.

**Figure 2 microorganisms-10-02107-f002:**
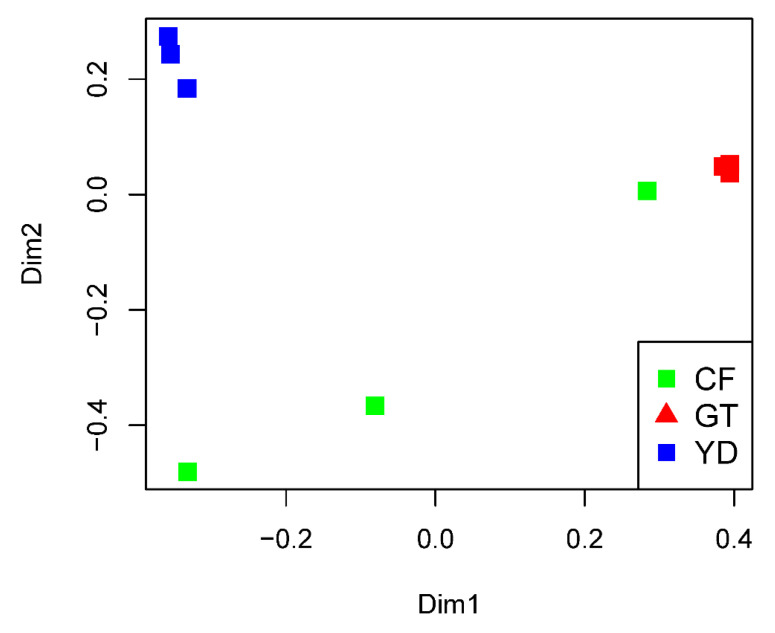
PCoA plot showing Bray–Curtis dissimilarities between the studied samples. GT: *G. tokunagai*; CF: *C. flaviplumus*; YD: Yeon Deung water.

**Figure 3 microorganisms-10-02107-f003:**
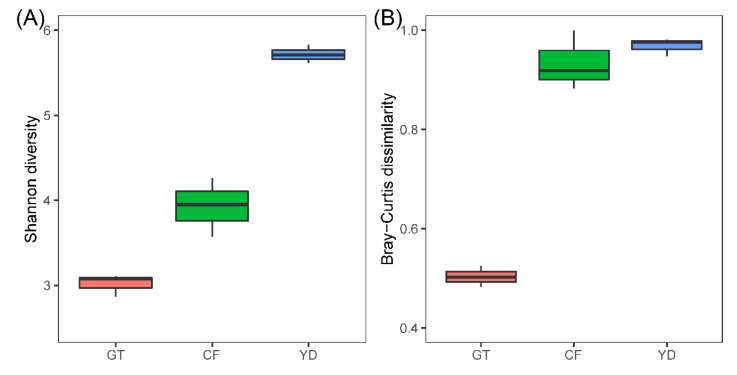
Shannon diversity (α-diversity) (**A**) and the within-group Bray–Curtis dissimilarity (β-diversity) (**B**) of the studied samples. GT (red): *G. tokunagai*; CF (green): *C. flaviplumus*; YD (blue): Yeon Deung water.

**Figure 4 microorganisms-10-02107-f004:**
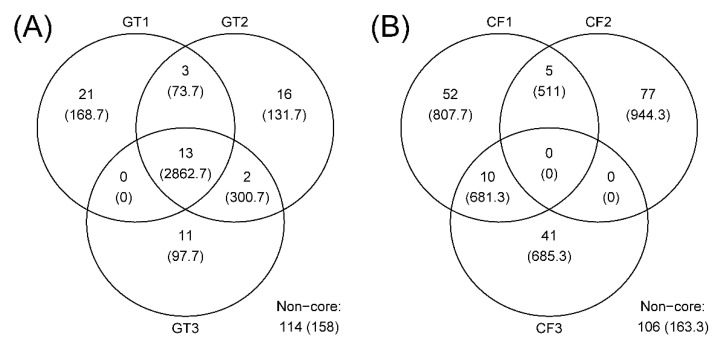
Venn diagrams showing the number of ASVs and their average abundance (the number of reads) in each chironomid species. (**A**) GT: *G. tokunagai*; (**B**) CF: *C. flaviplumus*.

**Figure 5 microorganisms-10-02107-f005:**
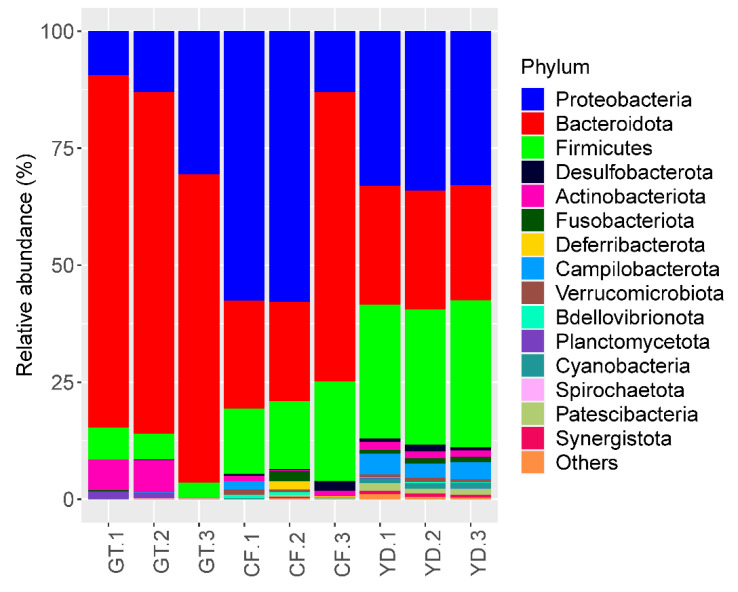
Relative abundance of the 10 most abundant phyla in the studied samples. GT: *G. tokunagai*; CF: *C. flaviplumus*; YD: Yeon Deung water.

**Figure 6 microorganisms-10-02107-f006:**
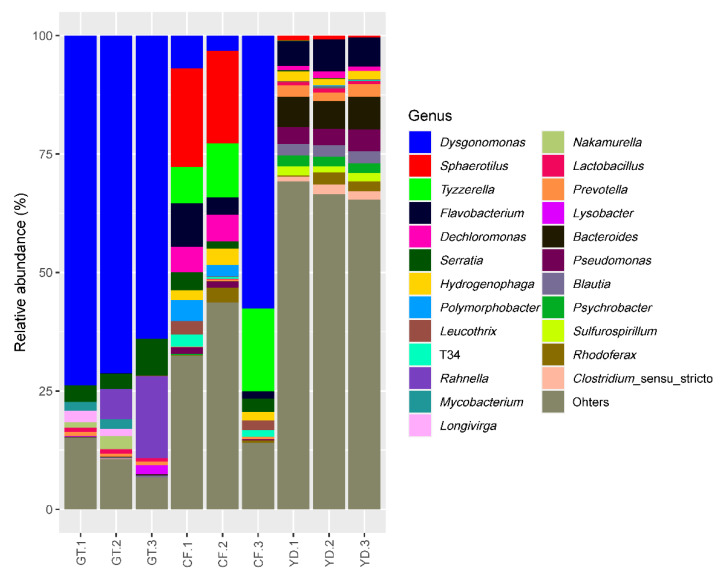
Relative abundance of the 10 most abundant genera in the studied samples. GT: *G. tokunagai*; CF: *C. flaviplumus*; YD: Yeon Deung stream water.

**Table 1 microorganisms-10-02107-t001:** Environmental conditions of the studied area.

**(*n* = 3)**						
**NO_2_ (μg/L)**	**NO_3_ (μg/L)**		**NH_4_ (μg/L)**		**TN (μg/L)**	
47.2 ± 0.1	4354.7 ± 15.7		557.2 ± 13.7		5607.8 ± 28.2	
PO_4_ (μg/L)	TP (μg/L)		SS (mg/L)		Chl-a (μg/L)	
112.6 ± 0.9	140.7 ± 2		2.6 ± 0.5		7.8 ± 0.6	
**(*n* = 7)**						
**Water Temperature (°C)**		**DO (mg/L)**	**SPC (μS/cm)**	**PSU**		**pH**
6.5 ± 0		16.4 ± 0.7	623.9 ± 3	0.3 ± 0		7.2 ± 0.2

**Table 2 microorganisms-10-02107-t002:** List of the KEGG gene categories related to resistance to antibiotics, toxic metals, and toxic compounds found in our study and in *C. ramosus* larval microbiome in Sele et al.’s study (2021). GT: *G. tokunagai*; CF: *C. flaviplumus*.

KEGG ID	Gene	Function	GT (%)	CF (%)
K07665	cusR, copR, silR	Two-component system, OmpR family, copper resistance phosphate regulon response regulator CusR	0.006	0.033
K03327	TC.MATE, SLC47A, norM, mdtK, dinF	Multidrug resistance protein, MATE family	0.072	0.047
K03297	emrE, qac, mmr, smr	Small multidrug resistance pump	0.011	0.019
K11741	sugE	Quaternary ammonium compound-resistance protein SugE	0.066	0.036
K11811	arsH	Arsenical resistance protein ArsH	0.001	0.014
K07245	pcoD	Copper resistance protein D	0.011	0.002
K07156	copC, pcoC	Copper resistance protein C	0.01	0.002
K07233	pcoB, copB	Copper resistance protein B	0.001	0.005
K02617	paaY	Phenylacetic acid degradation protein	0.001	0.003
K07803	zraP	Zinc resistance-associated protein	0	0.001
K15726	czcA	Cobalt–zinc–cadmium resistance protein CzcA	0.056	0.049
K16264	czcD, zitB	Cobalt–zinc–cadmium efflux system protein	0.066	0.031

## Data Availability

FASTQ Illumina sequence data have been deposited at the Sequence Read Archive (SRA) under project ID PRJNA880421.

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
