# Peer review of "Identification of Bacterial Communities in Laboratory-Adapted Glyptotendipes tokunagai and Wild-Stream-Inhabiting Chironomus flaviplumus"

_microorganisms, 2022, doi:10.3390/microorganisms10112107_

Round 1

Reviewer 1 Report

The article describes the gut microbiota of two chironomid species, one from the wild environment compared to another from breeding. It provides descriptive data on the gut microbiota composition of chironomid species and the potential influence of laboratory adaptation on the gut microbiota composition. However, some points need to be addressed prior publication of the article

Major

-        - As sequence number was normalized at a relatively low read number, please present the rarefaction curves to be sure that the observed effects are not associated to insufficient sequencing depth.

-       - Information about the host organisms are lacking. Please provide size range of the fourth instar larvae selected for the study as well as the methodology used to determine development stage.

-       -    Indicate Trophic state index calculation method in the material and method section

-       -  Please perform the analysis in order to indicate how different the gut microbiota of C. flaviplumus is different from the surrounding environment, especially as the DNA extract of the whole organism is performed, leading to the sample “contamination” with skin microbiota. If a procedure is applied to eliminate external microbiota, please describe it.

-        - While the authors use the Picrust analysis to emphasize on metal gene resistances. It would be strongly beneficial to indicate heavy-metal contamination levels of the sampling site.

-         - The English language need some adjustments

Minor :

- References are missing in the introduction section to support the author statements. Please provide them at the following lines:

-          L38-39 

-          L40 

-          L58 

- Please indicate the p-value in the manuscript when presenting the results to better show the levels of significant differences.

Reviewer 2 Report

Authors compared microbiomes from two different species (GT and CF) of Chironomidae which were often used for indicators for water quality.  It looked obscure why they used two different species,  one from lab-controlled environment and the other from wild environment.  Assuming that they probably wanted to compared microbiomes two different water quality, then water test from lab was missing in the manuscript.  For main result in the manuscript, I found inconsistency between statistics results and microbiome abundance data.  NMDS plot showed that the GT had almost no variation within a group so that all samples in this group were placed overlapped in the plot.  However I can easily see that Fig 3 and 4  showed  some variation even within a group even in phylum level.  GT3, and CH3 (btw, CF seems right not CH) were different from the rest of samples in the same group at phylum level.  GT1 and CH3 showed the variation with the group at genus level too. YD water samples were very similar across the samples so they should be placed very closely at NMDS plots.  Authors need to indicate what information of systematic level and microbiome abundance information was used for NMDS plot and to make sure if they miss any information. 

There is some minor comments.

Line15-16:  it was confusing the Cf was collected from tap water or captured. ‘leaking from tap water in 2020 and 2022” is not necessary. 

Line20. Uisng “For example” is not appropriate to be described in Abstract. 

Line 75.  No reason to be started in new paragraphs.  Need to move up to end of Lin74. 

Line 162.  Need to change 3.2 to 3.3

Table 2 and Table 3.  P-values need to be adjusted. 

Table 4.  list the KEGG IDs with statistically significant.  P-values also need to be adjusted. 

Reviewer 3 Report

Song et al. analyze the microbiome of two chironomid species, one cultured in the laboratory and the other from the wild. They compare both microbiomes and see differences in distribution and species variety, some related to detoxification mechanisms.

The abstract presents the work performed, but a clear conclusion will help the reader to understand better the relevance of the work.

In the introduction, it would be adequate to include a reference concerning the life cycle of chironomids for readers unfamiliar with these insects. However, it is not clear the reason for including Chironomus flaviplumus in the study. The authors indicate that other studies in chironomids compare wild and cultured animals from the same species. However, they fail to explain why they use a different species from the wild. According to the introduction, Glyptotendipes tokuganai is distributed in East Asia. Why do the authors select other species? They explain that a few species have been studied, which could be a reason, but why did they not use both species in the studies with cultured and wild animals? In addition, it is unclear what it means that wild Chironomus flaviplumus leaked from tap water in 2020 and 2022. It should be explained better; how do the authors know they leaked?

In material and methods, it is unnecessary to include the title of the OECD test since they have a number assigned (OECD test 233). It is unclear what a replica is in the present study. For GT, are each replicate coming from different cultures? Were they taken on different days or on the same day? It is the same problem for CH since it is not indicated if they were taken on different days or at different moments of the day. On the other hand, the authors filtered the water collected from the river with 0.22 micrometers filters. Usually, it would retain most of the bacteria, so what did they use to analyze the microbiome of the environment?

DNA extraction methodology is unclear. The authors explain that the DNA was extracted from the animals' whole bodies and the river's water (it is understood that the filtered water). They have used a kit. Then they say, “The total DNA from each filter was extracted using….”. Where did the filter come from? Is it the filter from the water filtering? Have the authors homogenized the animals? If so, which buffer have they used?

The discussion fits the results. However, it is unclear how comparing two species in different starting conditions could help to understand the diversity and the impact of the cultured vs. wild comparison. The bacterial diversity could also come from different feeding. The main weakness of the study is the absence of a proper reference to compare each population since there are too many variables involved that can justify the differences between them. In addition, using two species does not justify the relevance of the results of chironomids since they can differ depending on the environment. The chironomids are distributed in very different habitats: from Antarctica (Belgica antarctica) to desert (Polypedilum vanderplanckii), including those found in tropical areas (Chironomus sancticaroli), glaciers (Diamesa branickii), etc. So, the article requires major modifications or modifying the perspective focusing on the microorganisms.

Minor:

Line 73: “M4 median” should be substituted by “M4 medium”.

Round 2

Reviewer 1 Report

The authors responded to my comments in a satisfying way.

Reviewer 3 Report

The authors have modified the article following the recommendations of the reviewers.